# Rest to Resist: How Recovery Shields Well-Being from Work–Family Strain

**DOI:** 10.3390/bs15081089

**Published:** 2025-08-12

**Authors:** Cátia Sousa, Laura Silva

**Affiliations:** 1Faculdade de Ciências Humanas e Sociais, Universidade do Algarve, 8005-139 Faro, Portugal; a47730@ualg.pt; 2Centro Universitário de Investigação em Psicologia (CUIP), Universidade do Algarve, 8005-139 Faro, Portugal

**Keywords:** work–family conflict, recovery experiences, well-being, occupational stress, employee mental health

## Abstract

This study examines the mediating role of recovery experiences in the relationship between work–family conflict (WFC) and employee well-being. While WFC has been consistently linked to negative outcomes such as psychological distress and reduced life satisfaction, the mechanisms that may buffer its effects remain underexplored. Drawing on the Conservation of Resources Theory and the Effort-Recovery Model, we investigated whether four types of recovery experiences—psychological detachment, relaxation, mastery, and control—mediate the WFC–well-being relationship. A cross-sectional survey was conducted with 240 employees using validated self-report instruments. Data were analysed through correlation, regression, and mediation techniques, including bootstrapping procedures via PROCESS. The results confirmed a significant negative association between WFC and well-being. All four recovery experiences were positively related to well-being, with relaxation emerging as the strongest predictor. Mediation analyses showed that each of the recovery experiences partially mediated the relationship between WFC and well-being. These findings highlight the importance of recovery as a psychological buffer in the context of elevated work–family interference. Organizational practices that foster recovery—such as encouraging psychological detachment, offering flexible schedules, and promoting restorative activities—may contribute to sustaining employee mental health and resilience.

## 1. Introduction

In recent decades, profound societal and organizational changes—such as the increased participation of women in the workforce, the rise of dual-career households, and the growing permeability between work and family domains—have intensified the challenge of balancing professional and personal responsibilities. These transformations have contributed to a rise in work–family conflict (WFC), conceptualized as a form of inter-role conflict in which demands from the work and family domains are mutually incompatible ([15]). Extensive research has shown that WFC is associated with negative outcomes for employees, including poorer physical and psychological health, lower job satisfaction, and diminished overall well-being ([2]; [4]). Despite these well-documented consequences, the mechanisms that may buffer the adverse effects of WFC on employee well-being remain relatively underexplored. In particular, the potential protective role of recovery experiences—defined as psychological processes that help restore depleted personal resources ([46])—warrants further investigation. Recovery experiences such as psychological detachment, relaxation, mastery, and control have been linked to positive emotional states, reduced stress levels, and improved health outcomes ([44]; [6]). However, few studies have examined whether these experiences act as mediators in the relationship between WFC and well-being. Gaining insight into these mediating processes can enhance our understanding of how individuals may safeguard their mental health amid increasing work–family demands.

This study addresses this gap by examining the mediating role of recovery experiences in the relationship between work–family conflict and employee well-being in a Portuguese working population. Drawing on the Conservation of Resources Theory ([19]) and the Effort-Recovery Model ([32]), we propose that recovery experiences function as psychological buffers that attenuate the detrimental impact of WFC on well-being. By exploring these mechanisms, this research contributes to a more nuanced understanding of the dynamics between work and family roles, offering practical implications for organizations aiming to enhance employee well-being through recovery-supportive practices.

### 1.1. Work–Family Conflict and Employee Well-Being

Ongoing social and organizational transformations have increasingly challenged individuals in their efforts to balance work and family demands. Developments such as the rise of dual-career couples, single-parent households, and the pervasive influence of digital technologies have intensified the overlap between professional and personal spheres, creating a complex and dynamic interface ([28]; [17]). Work–family conflict (WFC) arises from these tensions and is commonly defined as a form of inter-role conflict in which the demands of work and family roles are mutually incompatible, such that participation in one role hampers engagement in the other ([15]). This conflict is bidirectional, encompassing work-to-family conflict (W → F), where work interferes with family responsibilities, and family-to-work conflict (F → W), where family obligations disrupt work performance ([13]).

Several theoretical frameworks help to explain the interplay between work and family domains. [54] ([54]) proposed the hypotheses of spillover, compensation, and segmentation. [19]’s ([19]) Conservation of Resources Theory highlights the finite nature of individual resources and the stress that arises when these are threatened or depleted. [23] ([23]) conceptualized WFC as a form of inter-role strain resulting from incompatible expectations. [33] ([33]) developed an integrative model categorizing WFC antecedents into work-related, family-related, and personality-related factors. Work-related antecedents include role stressors (e.g., overload, ambiguity), time demands, and limited job autonomy or flexibility. Family-related antecedents include parenting responsibilities, time pressure, and lack of family support. In terms of personality, traits such as neuroticism tend to exacerbate WFC, whereas conscientiousness and extraversion may serve as protective factors ([50]).

The consequences of WFC are far-reaching and span multiple life domains. At the individual level, WFC is associated with depression, anxiety, reduced life satisfaction, burnout, and physical health issues such as musculoskeletal disorders ([2]; [4]; [52]). It can also strain family relationships, increase parental stress, and negatively impact child development ([5]; [8]). In organizational contexts, WFC is linked to diminished job performance, increased absenteeism, lower affective commitment, and heightened turnover intentions ([1]; [3]).

Given these outcomes, recent research has increasingly focused on identifying protective mechanisms. One promising line of inquiry involves recovery experiences—psychological processes that help individuals replenish the resources depleted by the demands of work and family life. These experiences not only promote resilience and well-being but may also serve as crucial mediators in the relationship between WFC and employee well-being ([42]).

Given these outcomes, recent research has increasingly focused on identifying protective mechanisms. One promising line of inquiry involves recovery experiences—psychological processes that help individuals replenish the resources depleted by the demands of work and family life. These experiences not only promote resilience and well-being but may also serve as crucial mediators in the relationship between WFC and employee well-being. Importantly, empirical studies have shown that recovery experiences play a critical role in supporting employee well-being. For instance, psychological detachment from work during leisure time has been found to buffer the negative effects of workplace stress on well-being ([48]). [42] ([42]) demonstrated that recovery experiences mediated the relationship between WFC and well-being among female employees, mitigating psychological strain and enhancing life satisfaction.

This study seeks to examine how recovery experiences—specifically psychological detachment, relaxation, mastery, and control—mediate the relationship between work–family conflict and employee well-being. By doing so, it contributes to a deeper understanding of the psychosocial processes that support resilience in demanding work environments and provides evidence-based insights for human resource and organizational practices.

### 1.2. Recovery Experiences as Protective Mechanisms

Recovery experiences refer to the processes by which the functional systems, activated in response to stress, return to their baseline, pre-stressor levels ([44]). They can be conceptualized in three ways: situationally (the contexts in which recovery occurs), as a process (the mechanisms—such as activities or experiences—that facilitate detachment and restoration), and as an outcome (the emergence of positive affective and physiological states) ([47]). In this study, we focus on recovery as a psychological process.

Recovery experiences are distinct from recovery activities. While activities refer to what individuals do during their leisure time (e.g., exercising, reading), recovery experiences refer to the internal psychological processes that these activities evoke (e.g., feeling detached from work or relaxed). [46] ([46]) identified four core recovery experiences:Psychological detachment: the cognitive and emotional disengagement from work during non-work time, which reduces the activation of work-related mental systems. This experience is positively associated with well-being and is especially effective in countering persistent negative work-related thoughts ([6]; [31]).Relaxation: a state of calm and tranquillity, characterized by low physiological activation and enhanced positive affect. It has been linked to job satisfaction and is considered a passive form of recovery ([22]).Mastery: the pursuit of challenging non-work activities that foster learning, skill development, and a sense of accomplishment. Although it requires effort, mastery experiences are strongly associated with personal growth and well-being ([26]; [56]).Control: the degree of autonomy that individuals have over how they spend their free time. While its effects on well-being tend to be more modest, control contributes to psychological health by enhancing self-efficacy and perceived agency ([12]).

These mechanisms are grounded in two theoretical frameworks: the Effort-Recovery Model ([32]) and the Conservation of Resources Theory ([19]). The Effort-Recovery Model posits that recovery is essential to prevent the accumulation of strain caused by ongoing work demands. Without sufficient recovery, physiological and psychological systems remain activated, increasing the risk of chronic health problems. The Conservation of Resources Theory suggests that stress results from the threat or loss of valued resources, and recovery serves to replenish or rebuild these personal reserves ([19]).

A growing body of empirical evidence supports the effectiveness of recovery experiences. For example, psychological detachment has consistently been linked to reduced stress and enhanced well-being ([53]). Meta-analyses confirm its protective role, particularly on days characterized by high job stress ([37]). Similarly, relaxation and control are associated with greater emotional balance, while mastery contributes to motivation and engagement ([6]; [56]).

Engagement in recovery activities—such as creative hobbies, social interactions, and physical exercise—has been shown to enhance recovery experiences the following day ([11]). These experiences have been associated with increased vitality, improved interpersonal relationships, enhanced job performance, and higher life satisfaction ([24]; [22]).

In summary, recovery experiences function as protective psychological mechanisms that support the restoration of resources depleted by work demands. By enhancing energy, emotional regulation, and cognitive functioning, these experiences are essential for preserving employee health and sustaining engagement and well-being.

### 1.3. Recovery as a Mediator

Empirical research suggests that recovery experiences not only directly enhance well-being but may also buffer the negative effects of job stressors, including work–family conflict (WFC) ([9]; [38]; [36]). These experiences contribute to the maintenance of psychological functioning under sustained demands, acting as mediators in the stress–well-being relationship ([42]; [25]). Recovery is particularly relevant in the context of WFC, where competing demands from work and family domains exhaust emotional and cognitive resources. The capacity to mentally detach from work, engage in restorative activities, and exert control over one’s leisure time supports emotional regulation and helps reduce psychological strain.

Partial mediation is theoretically supported by the Conservation of Resources Theory, which acknowledges that resource loss may not be fully offset by recovery efforts. This perspective implies that, although recovery experiences can alleviate the detrimental effects of WFC, they may not eliminate them. Empirical studies by [9] ([9]), [55] ([55]), and [42] ([42]) have consistently demonstrated significant—but partial—mediation effects, emphasizing the complementary, rather than substitutive, role of recovery processes.

Drawing on theoretical frameworks and prior empirical findings, the present study proposes the following hypotheses:

**H1.** 
*Work–family conflict is negatively associated with employee well-being.*


Higher levels of WFC are expected to correlate with lower levels of psychological and emotional well-being, including reduced life satisfaction and increased stress-related symptoms.

**H2.** 
*Recovery experiences are positively associated with employee well-being.*


Experiences such as psychological detachment, relaxation, mastery, and control are anticipated to enhance well-being by promoting emotional regulation, restoring personal resources, and facilitating cognitive renewal.

**H3.** 
*Recovery experiences partially mediate the relationship between work–family conflict and employee well-being.*


That is, while the presence of recovery experiences may reduce the adverse effects of WFC, they are unlikely to fully neutralize them, instead acting as protective mechanisms that support employee resilience and psychological functioning.

These hypotheses aim to illuminate the underlying mechanisms through which recovery shapes the relationship between work–family demands and individual well-being. By doing so, the study provides evidence-based insights to inform organizational interventions and human resource practices that promote sustainable employee health and performance.

## 2. Materials and Methods

### 2.1. Participants

The study sample comprised 240 Portuguese working adults, the majority of whom identified as female (72.1%, *n* = 173), with 27.9% (*n* = 67) identifying as male. Participants’ ages ranged from 23 to 66 years (M = 43.45, SD = 9.80). Regarding family status, 32.5% reported having no children, 28.7% had one child, 32.1% had two children, and 6.7% had three or more. Most participants were married or in a domestic partnership (60.4%), followed by those who were single (27.9%) and divorced (11.7%).

Educational levels varied: 30.8% held a bachelor’s degree, 28.3% a master’s degree, 12.9% a postgraduate diploma, and 3.8% a PhD. The majority (58.3%) were employed in the public sector, 36.7% in the private sector, and 5% in mixed (public–private) institutions. Most participants worked in intellectual and scientific professions (61.3%), followed by mid-level technicians (15.8%) and administrative staff (8.3%). A large proportion were employed full-time (94.6%), and 75% held permanent contracts.

Inclusion criteria required participants to be at least 18 years old and currently engaged in paid employment, either full-time or part-time. Individuals who were unemployed, retired, or otherwise inactive in the labour market were excluded to ensure that all respondents could meaningfully reflect on work–family dynamics and recovery experiences.

A non-probability convenience sampling strategy was employed. Although no a priori power analysis was conducted, the final sample size (*n* = 240) exceeded the minimum recommendations for mediation analysis with bootstrapping procedures, as suggested by [18] ([18]).

### 2.2. Measures

Work–Family Conflict (WFC) was assessed using the Portuguese version of the Work–Family Conflict and Family–Work Conflict Scale ([40]), originally developed by [35] ([35]). The scale consists of 10 items rated on a 7-point Likert scale (1 = strongly disagree, 7 = strongly agree), comprising two subscales: work-to-family conflict and family-to-work conflict. In the present study, the internal consistency was high, with Cronbach’s alpha of 0.87 for the total scale and 0.86 for both subdimensions.

Recovery Experiences—Recovery was measured using the Portuguese adaptation of the Recovery Experiences Questionnaire ([46]; [29]), which includes 16 items divided into four subscales: psychological detachment, relaxation, mastery, and control. Responses were given on a 5-point Likert scale (1 = strongly disagree, 5 = strongly agree). In this study, the subscales showed acceptable to excellent internal consistency: α = 0.80 (detachment), α = 0.90 (relaxation), α = 0.89 (mastery), and α = 0.70 (control).

Well-Being—Well-being was assessed using the General Health Questionnaire—12 items (GHQ-12) ([14]), in the Portuguese version by [7] ([7]). This is a unidimensional measure of general mental health. Items were rated on a 7-point Likert scale (1 = strongly disagree, 7 = strongly agree), with higher scores indicating lower levels of psychological well-being. In the present study, the GHQ-12 demonstrated good internal consistency (α = 0.88).

### 2.3. Procedure

This was a cross-sectional, correlational study based on quantitative data collected through an online questionnaire. The survey was disseminated via social media platforms and professional networks between October 2023 and April 2024. Participants were informed of the study’s objectives and provided informed consent prior to participation. Anonymity and confidentiality were assured throughout. As the survey was distributed through open online channels, it was not possible to calculate an exact response rate.

Although demographic data such as age, gender, and employment sector were collected and reported, no statistical controls for potential confounding variables were applied in the main analyses. This decision reflects the exploratory nature of the study and is acknowledged as a methodological limitation.

The study was conducted within a post-positivist research framework, assuming that psychological constructs such as work–family conflict and recovery can be reliably measured and analysed through structured self-report instruments, while also recognizing the influence of contextual and subjective factors on interpretation.

### 2.4. Data Analysis

Descriptive statistics, Pearson correlations, and regression analyses were conducted using IBM SPSS Statistics version 28. Mediation analysis was performed using PROCESS macro (Model 4) developed by Hayes. The significance of indirect effects was tested via bootstrapping with 5000 samples. Correlation coefficients were interpreted according to [30] ([30]): values below 0.25 were considered weak, between 0.25 and 0.50 moderate, and above 0.50 strong.

## 3. Results

### 3.1. Descriptive Statistics

Table 1 presents the means and standard deviations for the key study variables. On average, participants reported moderate levels of work–family conflict (WFC; M = 3.06, SD = 1.19), with higher levels of work-to-family conflict (W → F; M = 3.69, SD = 1.48) compared to family-to-work conflict (F → W; M = 2.42, SD = 1.31), indicating greater perceived interference of work in the family domain.

Among the recovery experiences, mastery showed the highest mean (M = 3.74, SD = 0.96), followed by control (M = 3.44, SD = 0.85), relaxation (M = 3.43, SD = 1.09), and psychological detachment (M = 3.15, SD = 0.90). The highest overall mean was observed for well-being (M = 5.01, SD = 1.09); however, this must be interpreted in reverse, as higher scores on the GHQ-12 indicate lower psychological well-being.

To further contextualize the findings, cut-off analyses were conducted to identify participants with elevated levels of WFC and reduced well-being. Results showed that 42.9% of participants reported WFC scores above 4 (on a 1–7 scale), indicating substantial interference of work in family life. In contrast, only 10.0% reported FWC scores above 4, reflecting relatively low levels of family-related interference in work.

Regarding well-being, measured by the GHQ-12 where higher scores denote greater psychological distress, 53.3% of participants scored above the clinical threshold of 5, suggesting that over half of the sample experienced elevated levels of psychological distress.

### 3.2. Correlational Analysis

Pearson correlations revealed statistically significant relationships among the main study variables (Table 2). Work–family conflict (WFC) was negatively associated with several variables, showing the following:A moderate negative correlation with well-being (r = −0.432, *p* < 0.001) and relaxation (r = −0.294, *p* < 0.001),Weaker negative correlations with psychological detachment (r = −0.195, *p* = 0.002) and control (r = −0.128, *p* = 0.047).

All four recovery experiences were positively correlated with well-being, with the strongest association observed for relaxation (r = 0.461, *p* < 0.001), followed by control (r = 0.313, *p* < 0.001), mastery (r = 0.301, *p* < 0.001), and psychological detachment (r = 0.257, *p* < 0.001).

### 3.3. Regression Analyses

To test Hypothesis 1 (H1), a simple linear regression analysis showed that work–family conflict (WFC) significantly predicted lower levels of well-being, accounting for 18.7% of the variance (β = −0.432, *p* < 0.001). Both subdimensions—work-to-family conflict and family-to-work conflict—were also significant predictors of well-being (β = −0.384 and −0.350, respectively; *p* < 0.001).

To test Hypothesis 2 (H2), each recovery experience was entered separately into a simple regression model predicting well-being. The results indicated the following:Relaxation explained 21.3% of the variance (β = 0.461, *p* < 0.001)Mastery explained 9.0% (β = 0.301, *p* < 0.001)Psychological detachment explained 6.6% (β = 0.257, *p* < 0.001)Control explained 9.8% (β = 0.313, *p* < 0.001).

These findings support H2, confirming that each of the four recovery experiences is a significant positive predictor of well-being.

### 3.4. Mediation Analysis

To test Hypothesis 3 (H3), a parallel mediation model was estimated using PROCESS Model 4 ([18]). Work–family conflict (WFC) was entered as the independent variable, well-being as the dependent variable, and the four recovery experiences—psychological detachment, relaxation, mastery, and control—as parallel mediators. The total effect of WFC on well-being was significant (β = −0.432, *p* < 0.001). When the mediators were included in the model, the direct effect of WFC on well-being was reduced but remained statistically significant (β = −0.221, *p* < 0.001), indicating partial mediation. This suggests that a portion of the negative impact of WFC on well-being is accounted for by lower engagement in recovery experiences, although a substantial direct effect persists. The indirect effects were significant for the following:Relaxation (β = −0.111, 95% CI [−0.170, −0.059]),Control (β = −0.071, 95% CI [−0.125, −0.030]),Mastery (β = −0.066, 95% CI [−0.113, −0.028]),Detachment (β = −0.033, 95% CI [−0.065, −0.008]).

These findings indicate that all four recovery experiences significantly mediated the relationship between work–family conflict (WFC) and well-being, with relaxation demonstrating the strongest effect. Although nearly 43% of participants reported elevated levels of WFC, the mediation model was applied to the full sample rather than being limited to individuals experiencing high conflict. This decision is consistent with previous research (e.g., [42]; [48]), which supports examining recovery experiences as protective mechanisms across the general working population.

By treating WFC as a continuous predictor, the analysis captures variability in conflict intensity and allows the model to reflect more nuanced effects. This approach enhances both the generalizability and the practical applicability of the findings, offering insights that are relevant not only for high-risk groups but also for broader workforce interventions.

## 4. Discussion

This study aimed to investigate the mediating role of recovery experiences in the relationship between work–family conflict (WFC) and employee well-being. The results provide strong support for the proposed model, aligning with prior research and extending current knowledge in several meaningful ways.

As hypothesized, WFC was significantly and negatively associated with employee well-being, supporting H1. This finding is consistent with extensive literature linking inter-role conflict to increased psychological distress and decreased life satisfaction ([2]; [4]; [10]; [52]). Notably, participants reported greater work-to-family interference than the reverse, highlighting the persistent impact of professional demands on personal life. This asymmetry is in line with findings from [34] ([34]) and [27] ([27]), which indicate that work responsibilities tend to encroach more heavily on family obligations.

All four recovery experiences—relaxation, psychological detachment, mastery, and control—were positively associated with well-being, confirming H2. Among them, relaxation emerged as the strongest predictor, reinforcing the importance of low-arousal, pleasurable states for psychological restoration ([46]; [43]). Although detachment and control also showed significant relationships with well-being, their effects were weaker than anticipated. Previous studies (e.g., [6]; [53]) reported stronger associations, suggesting that the impact of these experiences may be context-dependent or moderated by individual and organizational factors ([51]).

Crucially, the mediation analysis confirmed H3: recovery experiences partially mediated the relationship between WFC and well-being. This indicates that while WFC has a direct detrimental effect on well-being, individuals who engage in restorative activities may be able to mitigate some of these negative outcomes. All four recovery experiences served as significant mediators, with relaxation again demonstrating the strongest indirect effect. These findings underscore the distinctive protective role of relaxation, consistent with [19]’s ([19], [20]) Conservation of Resources Theory and [32]’s ([32]) Effort-Recovery Model.

The partial nature of the mediation suggests that recovery alone cannot fully buffer the adverse effects of WFC. Although all four strategies contributed to mitigating the impact of WFC, their protective power was limited. This implies the likely involvement of other psychological mechanisms—such as coping strategies, emotional regulation, or perceived social support—in preserving well-being ([49]; [45]). These findings are consistent with prior studies reporting partial mediation (e.g., [16]; [41]) and emphasize the need for structural and policy-level interventions to complement individual recovery strategies.

Given that 42.9% of participants reported high levels of WFC and over half (53.3%) exceeded the GHQ-12 clinical threshold, promoting recovery through occupational health initiatives is not only beneficial—it is imperative.

### 4.1. Theoretical and Practical Implications

The study reinforces the theoretical proposition that recovery experiences serve as vital psychological mechanisms for restoring well-being in the face of inter-role conflict. It confirms that the relationship between work–family conflict (WFC) and well-being is not solely direct, but also operates through underlying restorative processes, thereby enriching our understanding of the work–family interface.

From a practical standpoint, these findings suggest that organizations can play a pivotal role in promoting employee well-being by fostering recovery-supportive workplace cultures. Specific interventions may include the following:Promoting psychological detachment through boundary management policies (e.g., discouraging after-hours emails)Facilitating relaxation by encouraging breaks, vacations, and stress reduction programsSupporting mastery by providing opportunities for non-work-related training or creative activitiesEnhancing control by offering flexible work schedules and greater autonomy over leisure time.

In addition to general initiatives, organizations should implement targeted programs that account for individual differences in recovery preferences, resources, and constraints. Examples include promoting recovery-supportive leadership styles, offering personalized work schedules, and incorporating recovery-oriented training modules focused on mindfulness, emotional regulation, or time autonomy ([12]; [21]). Human resource policies may also integrate recovery planning into career development and performance management frameworks. Occupational health programs should expand beyond traditional stress management approaches to include structured modules on relaxation, active leisure, and boundary-setting techniques, thus embedding recovery into the organization’s broader well-being strategy.

By cultivating environments that empower employees to engage in meaningful recovery experiences, organizations may effectively mitigate the negative impacts of work–family conflict and foster a healthier, more resilient workforce.

### 4.2. Limitations and Future Directions

This study is not without limitations. First, its cross-sectional design restricts the ability to draw causal inferences about the relationships among work–family conflict (WFC), recovery experiences, and well-being. Although the statistical mediation analyses suggest directional pathways, only longitudinal or experimental research can confirm causality. Second, data collection relied exclusively on self-report measures and a non-probability convenience sample. These methodological choices may have introduced biases such as social desirability and common method variance ([39]), while also limiting the representativeness of the sample and the generalizability of the findings to the broader working population. Third, although 42.9% of participants reported high levels of WFC and 53.3% scored above the clinical threshold for psychological distress on the GHQ-12, the mediation model was applied to the entire sample. This decision was theoretically justified to capture the full variability of WFC as a continuous predictor and to enhance generalizability ([42]; [48]). Fourth, cultural factors may limit the applicability of the findings beyond the Portuguese context. Work–family dynamics are influenced by societal norms and labour market conditions; therefore, replication in cross-cultural samples is recommended. Future studies could examine moderating variables such as gender, parental status, job type, or work schedule flexibility to better identify for whom recovery is most effective. Additionally, qualitative or mixed-method designs may offer deeper insights into the subjective experience of recovery. Fifth, although demographic characteristics such as age, gender, and employment sector were recorded, potential confounding variables were not statistically controlled in the analyses. This limits the ability to isolate the unique contribution of each predictor and raises the possibility that unmeasured factors—such as caregiving responsibilities, workload intensity, or organizational support—may have influenced the observed relationships. Future research should incorporate such variables into multivariate models to account for their potential effects.

Finally, future research should develop, and test interventions aimed at enhancing recovery opportunities both at work and at home. Exploring how digital communication, remote work arrangements, or organizational support policies influence recovery effectiveness may yield valuable insights for designing healthier and more sustainable work environments.

In summary, this study contributes to the growing body of literature by clarifying the role of recovery experiences in protecting employee well-being from the negative effects of work–family conflict. It underscores the importance of supporting employees’ recovery processes—particularly relaxation—as a practical and evidence-based strategy to foster psychological resilience in modern workplaces.

## 5. Conclusions

This study contributes to the growing body of literature on work–family dynamics by providing empirical evidence that recovery experiences—specifically psychological detachment, relaxation, mastery, and control—partially mediate the negative relationship between work–family conflict (WFC) and employee well-being. These findings underscore the importance of recovery processes as psychological resources that enable individuals to cope with the strain generated by conflicting work and family demands.

By integrating the Effort-Recovery Model and the Conservation of Resources Theory, this research highlights the complex and multifaceted ways in which individuals restore depleted energy and safeguard their psychological health. Recovery is not merely a passive consequence of time away from work but an active and dynamic process that requires awareness, opportunity, and support.

The results demonstrate that although WFC compromises well-being, employees who engage in effective recovery strategies are better equipped to maintain psychological functioning and resilience. This has important practical implications for organizations aiming to promote sustainable employee well-being and performance. Employers should not only address the sources of conflict—such as excessive workloads or inflexible schedules—but also cultivate environments that actively support recovery. This includes promoting flexibility, autonomy, boundary management, and the meaningful use of leisure time.

In an era defined by increasingly blurred boundaries between work and personal life—especially within hybrid and remote work contexts—fostering a culture that supports recovery is no longer optional; it is a strategic imperative for individual health and organizational sustainability. Future policies and interventions should be guided by this understanding, ensuring that recovery is embedded into the fabric of organizational life.

## Figures and Tables

**Table 1 behavsci-15-01089-t001:** Means and standard deviations of the variables under study.

Variables	M	SD
WFC-FWC	3.06	1.19
WFC	3.69	1.48
FWC	2.42	1.31
Relaxation	3.43	1.09
Mastery	3.74	0.96
Detachment	3.15	0.90
Control	3.44	0.85
Well-Being	5.01	1.09

**Table 2 behavsci-15-01089-t002:** Correlations.

	1	1.1	1.2	2.1	2.2	2.3	2.4
1. WFC-FWC	1						
1.1 WFC	0.870 **	1					
1.2 FWC	0.830 **	0.447 **	1				
2.1 Relaxation	−0.294 **	−0.326 **	−0.165 *	1			
2.2 Mastery	−0.108	−0.087	−0.097	0.456 **	1		
2.3 Detachment	0.195 **	−0.180 **	−0.149 *	0.476 **	0.121	1	
2.4 Control	−0.128*	−0.112	−0.106	0.405 **	0.355 **	0.255 **	1
3. Well-Being	−0.432 **	−0.384 **	−0.350 **	0.461 **	0.301 **	0.257 **	0.313 **

* *p* < 0.05; ** *p* < 0.01.

## Data Availability

The original data presented in this study are openly available in OSF at https://doi.org/10.17605/OSF.IO/27A8E (accessed on 1 August 2025).

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
