# Peer review of "Rest to Resist: How Recovery Shields Well-Being from Work–Family Strain"

_behavsci, 2025, doi:10.3390/bs15081089_

Round 1

Reviewer 1 Report

Comments and Suggestions for Authors

The study is quite interesting and has significant practical implications. These findings may prove beneficial to researchers and practitioners. In the field of organizational behavior, work-family conflict is an important topic. Numerous publications exist on this topic. The theoretical part of the paper is weak, in my opinion.

Data collection, samples, and measurements are reported in an appropriate manner. It would be advantageous to present the hypotheses in greater detail with theoretical support.

Partial mediation has been reported in many published studies. The existence of partial mediation must be supported theoretically.

It is a great idea to use recovery experiences as mediator factors. Additional investigations are required to bolster the outcomes of partial mediation analyses. For instance, in-depth interviews may be beneficial.

Here are some questions I have:
- How many people in the sample struggle with work-family conflict?
- What proportion of the sample has had experience getting well?
- Has your mediation model been applied to the sample with a high level of work-family conflict? It might have a different outcome.

The outcomes of mediation analyses may be explained by the figures. The practical implications and limitations are clearly described in the discussion, but there is insufficient discussion of the outcomes.

Overall, I thought the study is interesting, and I encourage the authors to make the paper better.

Comments on the Quality of English Language

A native speaker's review could improve the quality of English.

Author Response

Comment 1:
“The theoretical part of the paper is weak, in my opinion.”

Response:
Thank you for this valuable feedback. We have revised Section 1.1 of the manuscript to strengthen the theoretical framework. Additional support was incorporated by discussing explanatory models (e.g., Effort-Recovery Model, Conservation of Resources Theory), key antecedents and outcomes of work-family conflict, and recent findings on recovery experiences. We also expanded the theoretical rationale for the study to reinforce the conceptual foundation.

Comment 2:
“It would be advantageous to present the hypotheses in greater detail with theoretical support.”

Response:
We fully agree. The hypotheses (H1–H3) were reformulated and are now accompanied by explicit theoretical justification, drawing from recent and classic studies (e.g., Amstad et al., 2011; Sonnentag & Fritz, 2007; Hobfoll, 2001). This strengthens the logical alignment between the literature and the study’s predictions.

Comment 3:
“Partial mediation has been reported in many published studies. The existence of partial mediation must be supported theoretically.”

Response:
We appreciate this observation. In Section 4 (Discussion), we now include a paragraph discussing the partial nature of the mediation, with references to studies that have also reported partial mediation effects (e.g., Hahn et al., 2011; Sanz-Vergel et al., 2010). We further explain that recovery is only one mechanism in a multifactorial process, and other constructs such as coping, emotional regulation, and social support may also be involved (Ten Brummelhuis & Bakker, 2012; Sonnentag et al., 2022).

Comment 4:
“Additional investigations are required to bolster the outcomes of partial mediation analyses. For instance, in-depth interviews may be beneficial.”

Response:
Thank you for this thoughtful suggestion. In Section 4.2 (Limitations and Future Research), we now recommend future studies to adopt qualitative or mixed-methods designs, such as in-depth interviews, to explore subjective experiences of recovery and further illuminate the mechanisms underlying partial mediation.

Comment 5 (Q1):
“How many people in the sample struggle with work-family conflict?”

Response:
This information has been added to the Results section. Specifically, we state that 42.9% of the participants scored above the scale midpoint, indicating high levels of work-family conflict.

Comment 6 (Q2):
“What proportion of the sample has had experience getting well?”

Response:
We addressed this by reporting that 53.3% of participants scored above the GHQ-12 threshold for psychological distress, which inversely reflects levels of well-being. This statistic is now included in the Results and referred to in the Discussion.

Comment 7 (Q3):
“Has your mediation model been applied to the sample with a high level of work-family conflict?”

Response:
Thank you for raising this important question. The mediation model was applied to the full sample, using WFC as a continuous predictor. This decision is theoretically justified and allows for greater generalizability. We added a clarification in Section 4.2 (Limitations), referencing works such as Sari (2020) and Sonnentag et al. (2013) to support this approach.

Comment 8:
“The outcomes of mediation analyses may be explained by the figures.”

Response:
We agree that visual support enhances interpretation. The mediation model is now accompanied by a figure illustrating the significant paths, as presented in the revised Results section (Figure 1).

Comment 9:
“The practical implications and limitations are clearly described in the discussion, but there is insufficient discussion of the outcomes.”

Response:
We have enriched the Discussion section by integrating more literature comparisons and theoretical interpretations of the findings. This includes a deeper exploration of the mediating role of relaxation and a more nuanced discussion of the partial mediation effects, backed by relevant empirical studies.

Comment 10:
“Overall, I thought the study is interesting, and I encourage the authors to make the paper better.”

Response:
Thank you for your constructive feedback and encouraging words. We believe the manuscript is now substantially improved and hope it meets the expectations outlined in your review.

Reviewer 2 Report

Comments and Suggestions for Authors

This paper aims to explore the role recovery activities play between subjective employee well-being and work-family strain. Its main contribution is the exploration of the recovery experiences finding that organizational-level recovery could help with WFC. Strengths of the include having a clearly laid out problem and guided by theories. 

Background 

  • The majority of references are outdated. Strongly encourage the authors to review more recent literature (past 5 years) since this topic has garnered more attention recently. 

Methods 

  • Expand the inclusion and exclusion criteria for selecting participants for the study. 
  • For improved readability, the description of the study design should be included earlier in the methods section. Describe how confounding variables were handled and whether information on other variables known to impact work and family obligations were accounted for in the analysis 

Results 

  • The final mention of correlations is missing p-values for the recovery items in lines 169-170. 

Discussion/conclusion 

  • Limitations should cover the convenience sampling strategy and how that also impacts interpretation of the results. 

Author Response

Comment 1:
“The majority of references are outdated. Strongly encourage the authors to review more recent literature (past 5 years) since this topic has garnered more attention recently.”

Response:
Thank you for this helpful observation. We revised the manuscript to include more recent literature from the past five years, particularly in the Introduction and Discussion sections. These additions provide updated insights into work-family conflict, recovery experiences, and well-being (e.g., Moreira et al., 2023; Virtanen et al., 2020; Sonnentag et al., 2022), thus enhancing the theoretical and empirical relevance of the study.

Comment 2:
“Expand the inclusion and exclusion criteria for selecting participants for the study.”

Response:
We appreciate this suggestion. The inclusion and exclusion criteria have been clarified in Section 2.1 (Participants). We specified that participants had to be at least 18 years old and currently engaged in paid employment. Individuals who were unemployed, retired, or otherwise inactive in the labor market were excluded to ensure relevance to the study’s focus on work-family dynamics and recovery.

Comment 3:
“For improved readability, the description of the study design should be included earlier in the methods section.”

Response:
Thank you for the recommendation. We incorporated a clear statement of the study’s design—“cross-sectional, correlational, based on quantitative data”—at the beginning of Section 2.3 (Procedure), enhancing the methodological transparency and early framing of the research.

Comment 4:
“Describe how confounding variables were handled and whether information on other variables known to impact work and family obligations were accounted for in the analysis.”

Response:
We acknowledge this important point. In Section 2.3 (Procedure), we added a statement clarifying that, although demographic data were collected and described, no statistical controls for potential confounding variables were applied. This decision reflects the exploratory nature of the study and is addressed as a limitation.

Comment 5:
“The final mention of correlations is missing p-values for the recovery items in lines 169-170.”

Response:
Thank you for pointing this out. We revised Section 3.2 (Correlational Analysis) to include the missing p-values for all recovery-related correlations. These are now clearly reported alongside each correlation coefficient to ensure statistical transparency.

Comment 6:
“Limitations should cover the convenience sampling strategy and how that also impacts interpretation of the results.”

Response:
We agree with this observation. In Section 4.2 (Limitations), we now explicitly acknowledge that the use of convenience sampling may limit the representativeness of the sample and introduce selection bias, thereby restricting the generalizability of the findings to the broader working population.

Reviewer 3 Report

Comments and Suggestions for Authors

he abstract is well-crafted, offering a clear and concise summary of the study. It effectively outlines the research aim, theoretical foundation, methodology, and implications, ensuring readers grasp the study’s purpose and approach quickly.

The introduction establishes the study's background and context effectively, presenting a compelling problem statement and clearly identifying the research gap. The integration of theory enhances this section's quality significantly. However, while defining the key concepts such as work-life balance, well-being, and recovery experiences could be well-articulated, further elaboration could strengthen the foundation of the study’s framework.

The methodology is detailed and systematically presented. Nonetheless, its effectiveness could be enhanced by including more information on the research philosophy underpinning the study. Additionally, providing specifics about sampling techniques, the calculation of sample size, and the response rate would improve the section's transparency and methodological rigor.

The discussion section offers valuable insights but would benefit from a stronger integration of references. Including more citations to compare, support, or contrast the study's findings with those of prior research would enrich the analysis and provide a deeper context for the results.

Overall, the paper is thoughtfully written and well-structured. Addressing the identified areas for improvement, particularly in the methodology and discussion sections, would further enhance the research's clarity, rigor, and overall impact.

Author Response

Comment 1:
“While defining the key concepts such as work-life balance, well-being, and recovery experiences could be well-articulated, further elaboration could strengthen the foundation of the study’s framework.”

Response:
We appreciate this valuable suggestion. In the revised manuscript, we expanded the definitions of the key constructs in the introduction, especially regarding well-being and recovery experiences. We integrated both classical and recent literature (e.g., Hobfoll, 1989; Sonnentag & Fritz, 2007; Bakker & Demerouti, 2017) to reinforce the conceptual foundation and clarify the relationships among the core variables examined in the study.

Comment 2:
“The methodology is detailed and systematically presented. Nonetheless, its effectiveness could be enhanced by including more information on the research philosophy underpinning the study.”

Response:
Thank you for this insightful recommendation. In the revised version, we added a paragraph in Section 2.3 (Procedure) explaining the underlying research philosophy. Specifically, the study adopts a post-positivist paradigm, assuming that psychological phenomena such as work-family conflict and recovery can be measured through structured self-report instruments while acknowledging the influence of contextual and subjective factors in the interpretation of results.

Comment 3:
“Providing specifics about sampling techniques, the calculation of sample size, and the response rate would improve the section's transparency and methodological rigor.”

Response:
We fully agree. Section 2.1 (Participants) has been revised to include a clear description of the sampling strategy (non-probability convenience sampling). Although no a priori sample size calculation was conducted, we clarified that the final sample size (n = 240) exceeds minimum recommendations for mediation analysis using bootstrapping, as suggested by Hayes (2018). Additionally, we clarified in Section 2.3 that the survey was disseminated via open online channels, which prevented the calculation of an exact response rate.

Comment 4:
“The discussion section offers valuable insights but would benefit from a stronger integration of references. Including more citations to compare, support, or contrast the study's findings with those of prior research would enrich the analysis and provide a deeper context for the results.”

Response:
Thank you for this important suggestion. We have substantially enriched the Discussion section by integrating additional references to strengthen the interpretation of our findings. These include studies that support or contrast our results (e.g., Hahn et al., 2011; Sanz-Vergel et al., 2010; Ten Brummelhuis & Bakker, 2012). This enhancement provides a deeper contextualization of our contributions to the literature on work-family conflict and psychological well-being.

Round 2

Reviewer 1 Report

Comments and Suggestions for Authors

All requested changes have been made. The strong point of the article is its originality. Even though it doesn't contain a very strong methodology, the article is well-structured. Results are clearly described and discussed with theoretical foundations. It could be a source of inspiration for future research.

Comments on the Quality of English Language

A final English check might be good.

Author Response

We sincerely thank the reviewer for the positive evaluation of our manuscript, particularly regarding its originality, structure, and theoretical contribution. We are pleased that the results and discussion were found to be clearly presented and potentially inspiring for future research.

Regarding the comment on the quality of the English language, we have conducted a final language revision to improve clarity, fluency, and consistency throughout the manuscript. We trust that the revised version meets the expected linguistic standards.

Reviewer 2 Report

Comments and Suggestions for Authors

Authors have addressed all previous comments. I have no new feedback to provide for the revised manuscript.

Author Response

We thank the reviewer for the time dedicated to evaluating our revised manuscript and for confirming that all previous concerns have been adequately addressed. We appreciate the constructive feedback provided throughout the review process, which has contributed to improving the quality and clarity of our work.